# Afro-Latin@ Representation in Youth Literature: Affirming Afro-Latin@ Cultural Identity

Ada Malcioln Martin 

College of Education, University of Arizona, Tucson, AZ 85721, USA; ammart75@arizona.edu

**Abstract:** Studies show that diverse representation in children's literature can positively impact the self-perceptions of marginalized children. To promote feelings of self-worth, children must see their cultural identities authentically portrayed in a manner that does not promote stereotypes in stories that affirm and support their world experiences. This essay focuses specifically on Afro-Latin@ identity in the United States and the role Afro-Latin@ representation in children's and young adult literature can play in shaping Afro-Latin@ feelings regarding race and cultural heritage, and in constructing and affirming self-identity and feelings of self-worth.

**Keywords:** Afro-Latin@; mestizaje; blanqueamiento; Afro-Latinidad; colorismo

## 1. Positionality

From an emic perspective and cognizance of the inherent subjectivity in research, this author acknowledges how her cultural, personal, and professional experiences may impact this paper's objectivity. For this reason, the author wishes to disclose information about her cultural identity. The author is a cinnamon-skinned, cisgender, and middle-aged woman who presents phenotypically as Black and identifies culturally as Afro-Latina. Her father was a native Spanish speaker of Afro-Panamanian descent who immigrated to the US from Mexico in the 1940s. Her mother was of Afro-Caribbean heritage, born in the United States to Bajan/Barbadian parents (one of African descent, the other of African and Scottish ancestry). The author was born and raised in New York City during the early 70s and late 80s, mainly growing up in Harlem and in the predominantly [2]Latin@ neighborhood of Washington Heights. Her cultural and racial background undoubtedly informs this research as they position her as an insider in this community being discussed.

## 2. Introduction

Afro-Latin@[1] identity is complex as the mere presence of Afro-Latin@ populations in the Americas works to dismantle popular racialized narratives about who is Latin@ and who is not, particularly in the context of racial identification within the United States, where there tends to be a limited approach to capturing the fullness and rich diversity of Latin@ identity. Additionally, to be Black and forced to categorize oneself as simply Latin@ while denying racial or phenotypic makeup dismisses the unique challenges that Afro-Latin@s face within their respective communities, within the United States, and beyond (Adames et al. 2016; Comas-Diaz 2001). The purposeful erasure of Black identity, supported by the US and Latin American colonial machine, has led many Afro-Latin@s to distance themselves from their African origins. Historically, there have been strategic attempts throughout Latin America to minimize the presence of Black people and their contributions to Spanish-speaking societies.

In any discussion about the Afro-Latin@ experience, anti-Blackness and Black erasure must be considered if a complete picture of the challenges faced by these communities is to be entertained. Without a fundamental understanding of the historical treatment of these groups, it is difficult to comprehend the importance Afro-Latin@ representation

in children's literature plays in helping to positively shape the identities of Afro-Latin@ children. This essay identifies critical historical elements that have led to the erasure of the Afro-Latin@ identity within larger Latin@ communities and the role that erasure has played in formulating Afro-Latin@ perceptions around identity and belonging. Additionally, the author explores the necessity and importance of positive Afro-Latin@ representation within US youth literature in promoting cultural pride among young Afro-Latin@s in the United States.

## 3. Afro-Latin@ Identity

Per Pew Research, as of 2020 there are approximately 6 million adults in the United States that identify as Afro-Latin@, and they make up 2% of the US adult population and 12% of the adult Latin@ population (Gonzalez-Barrera 2022). Pew points to the difficulty associated with accurately representing the number of Afro-Latin@s in this country as much of the complexity is tied to the way the information has been historically captured by the US Census Bureau. The Pew Research Center took a different approach to census taking insofar that they moved away from the US Census construction of Afro-Latinidad as being those who identify as Hispanic and Black, to a more pointed approach in which they asked those surveyed whether they self-identify as Afro-Latin@ (Gonzalez-Barrera 2022). The 2020 census results showed that 1.2 million people (all ages) identified as Black and Hispanic while the Pew Research approach garnered a larger number of individuals who identify as Afro-Latin@ thus highlighting the complexity associated with Afro-Latin@ identity (Gonzalez-Barrera 2022). Per Gonzalez-Barrera (2022), Afro-Latin@ responses to Census Bureau questions regarding racial identity display vastly different responses as approximately three-in-ten Afro-Latin@s identified their race as white, 25% chose Black, 23% selected other, 16% chose the multiple race option, and 1% identified as Asian. Additionally, those Afro-Latin@s that did not consider themselves Latin@ were more likely to identify as Black (59% vs. 17%), which further contextualizes the deep divide among Afro-Latin@s in the ways in which they choose to identify.

When discussing Afro-Latin@ demographics, it is essential to clarify the usage of the terms "Latin@" and "Hispanic". The terms "Latin@" and "Hispanic" are often used interchangeably without acknowledging their distinct meanings (Rochin 2016). The term "Latin@" is derived from Latin and refers to those individuals who originate from Latin America despite their national ethnic, or linguistic backgrounds and can include those who originate from Brazil and the Indigenous people of Latin America (Rochin 2016). The origin of the term "Hispanic", comes from the term "Hispania" which references Spain; therefore, individuals with Spanish ancestry are the only ones that would be considered "Hispanic" (Rochin 2016). This grouping excludes those of Brazilian origin and those individuals from English- or French-speaking regions of Latin America (Rochin 2016). Comas-Diaz (2001) expressed that the term "Hispanic" in essence is an umbrella term generated by the United States Bureau of the Census to identify individuals of Spanish origin in the 1970 census. Comas-Diaz (2001) situates the usage of this terminology in relation to internalized colonization and suggests that its usage was promoted by conservative political actors who believed their European-Spanish ancestry to be superior to dominated groups. The important distinction to make in terms of this discussion is that all individuals of Latin American origin are Latin@, while not all Latin@s are Hispanic.

The Pew Research Center's attempt at capturing Afro-Latin@ identity has shown the inherent difficulty with the task. Afro-Latin@ presence in the United States is undeniable, but given the monolithic way that racial classification operates in this country, many are unsure of how to view identity that falls outside of US societal norms, leaving those in charge of racial enumeration confused about the complex nature of Afro-Latin@ identity and wondering what or who are Afro-Latin@s. The Afro-Latin@ Reader: History and Culture in the United States speaks of this confusion surrounding the term and points to the fact that most people have bought into the mutual exclusivity of Blackness, which defaults to a singular notion of what it means to be Black in the United States and is centered

around the Black and white binary. Blackness is defined through the lens of the African American or Southern Black experience and does not account for the complexity that exists within Afro-Latin@ or Afro-Caribbean identity. For the purpose of this discussion, we define *Afro-Latinidad* as those individuals of African descent who are from Mexico, Central and South America, and the Spanish-speaking Caribbean, and those individuals of African descent in the United States who are of Latin American and Caribbean origins (Jiménez Román and Flores 2010).

## 4. Anti-Black Colonial Narratives in Latin America and Beyond

In discussions surrounding Afro-Latin@ identity in Latin America and the United States, it is important to provide some historical context as to how Blackness operates within the confines of most Latin American countries. We must first acknowledge that the existence of Black people in Latin America is indisputable. From a diasporic perspective, the African presence is undeniably felt throughout Latin America and the Caribbean. Of the Africans forcibly brought to the Americas as part of the Middle Passage, approximately 10.7 million of the enslaved landed in Latin America and the Caribbean. At the same time, only 0.036 percent arrived in the United States (Hernández 2022). Much like in the United States, the after-effects of colonization and racial stratification in Latin America have created an environment where Afro-Latin@s, especially those with dark skin and African features, are often poor and socioeconomically disadvantaged as opposed to their white Latin@ counterparts (Hernández 2022). For example, this behavior can be seen in Cuba prior to the Cuban revolution of 1959, where dark-skinned Afro-Cubans were denied access to certain jobs that light-skinned mixed-race Cubans were allowed to occupy due to the general belief that the lighter-skinned applicants were racially superior (Hernández 2022). So, despite popular narratives often perpetuated within Latin@ communities about the absence of racism and racial discrimination in Latin America, historically speaking, just as in the US, racism is pervasive and a fact of life in Latin America and the Caribbean (Hernández 2022).

To situate the importance of positive Afro-Latin@ representation in children's literature, it is necessary to first identify how anti-Blackness in Latin America and beyond has historically worked to inform the attitudes and perceptions of Latin@s and Afro-Latin@s about Blackness and Black people. Discussions around anti-Blackness in many Latin@ communities are often situated within the racialized paradigm fostered in the United States that positions African Americans as racially inferior. Additionally, some Latin@ circles suggest that racial prejudice and stereotyping are uniquely US constructs not reproduced in Latin America. Bonilla-Silva (2004) outlined the tri-racial system that exists in Latin American countries that places white and light-skinned Latin@s and Asians at the top of the pecking order and dark-skinned Latin@s of all races at the bottom of the hierarchy. Perez Lopez (2017) affirmed that this stratification explains the perceptions of some Latin@ groups around the existence of racism in Latin America and the United States. Additionally, it provides insight into how racist incidents might possibly be disregarded or undocumented within Latin American countries: "This model explains how, for example, in Mexico it is normal/common to say that there is no race or racism. It is also common to say that in the United States people are very racist, but not in Mexico; and as a result, racism is scarcely documented in Mexico" (Perez Lopez 2017, p. 3).

Tanya Kateri Hernández (2022), a leading scholar of Afro-Latin@ experience and Anti-Blackness in the Latin@ communities, discusses the denial on the part of some Latin@s of the existence of racism against Afro-Latin@s in Latin America and within Latin@ communities. Hernández (2022) situates the root of this denial in the concept of mestizaje, which refers to the racial mixture present among the population within many Latin American countries. Mestizaje denies and counters narratives of racism as it enables Latin@s to claim racial innocence by suggesting that due to their own racial mixture, and the racial mixture that exists in many Latin@ families, that they, meaning Latin@ people, are somehow incapable of harboring Anti-Black sentiments or racist attitudes toward Afro-Latin@s or African Americans (Hanchard 1994; Costa Vargas 2004; Johnson 2020).

Quiros and Dawson (2013) also spoke of the existing racial hierarchy in the US and the historical construction of race having moved away from the idea of the one-drop rule, the Jim Crow era law that specified that anyone possessing Black ancestry must identify as Black, to more of a tri-racial system where those associated with whiteness (in this case "white" Latin@s) earn greater privilege and acceptance over darker-skinned Latin@s (Hickman 1996). It is important to note that the one-drop rule was not practiced in Latin America; as mentioned previously, the idea of mestizaje was supported, and the racial mixing of Europeans with non-Europeans was encouraged and used to promote colonial interests and to perpetuate white supremacy in Latin America during the Postcolonial era. The end goal of mestizaje was to "mejorar la Raza" or, as the English translation suggests, "better the race" by working strategically to actively dilute the existing non-European gene pool. This process was designed purposefully to eliminate Black and Indigenous populations from the larger population (Chavez-Dueñas et al. 2014; Charles 2021). The presence of existing racialized categories and attempts at social stratification within Latin America and the United States intentionally privilege light-skinned or white Latin@s over dark-skinned Afro-Latin@s, thus causing many Afro-Latin@s to suffer psychological and socioeconomic distress based upon these socially constructed and widely held perspectives (Hochschild and Weaver 2007; Bonilla-Silva 2004; Ore 2003).

The colonized notion of blanqueamiento, the belief that whiteness and one's approximation to whiteness are associated with intelligence, status, and wealth, while Blackness and Indigenousness are lacking in sophistication or refinement, is historically a commonly held proposition that permeates the Latin@ community and many Latin@ families today (Garcia 2015). The practice of colorism, the "process that privileges light-skinned people of color over dark in areas such as income, education, housing, and the marriage market", is common among Afro-Latin@s and other racialized groups (Hunter 2007, p. 237). However, it, too, is rooted in colonialism and white domination and promotes the idea that white European identity is ideal and preferential over Indigenous or African ancestry (Fanon 1952; Hunter 2002; Quiros and Dawson 2013). Additionally, colonization has contributed to perceptions about the undesirability of Blackness or Brownness by painting these groups as lazy and ugly (Fanon 1952; Hunter 2002; Quiros and Dawson 2013; Garcia 2015). This colonized belief system is reproduced in many Latin@ communities, and is often present particularly in discussions of idyllic beauty, preferred skin tone and hair texture, and in instances where one's societal positioning is dependent upon their proximity to whiteness (Quiros and Dawson 2013).

Despite the predominance of anti-Blackness in Latin America, its presentation does not emerge uniformly across all Latin@ communities as perceptions about Blackness and the willingness to align oneself with Blackness differ greatly depending upon the country of origin and an individual's proximity to Blackness. Johnson (2020) posits in his study examining the effects of racial fluidity on Black consciousness, that among mixed and Afro-Panamanians, phenotype is a better predictor of how an individual perceives things such as system equality and whether they believe in the possibility of Black mobilization. Those who appear more phenotypically Black (darker-skinned) tend to have stronger perceptions of systemic racial equality while those who appear more racially mixed tend to express a belief in collective Black mobilization to address systemic issues. Johnson (2020) also found in his discussion with Panamanian participants that mestizaje did not negatively impact the way mixed-race Panamanians perceive their Black identity. Surprisingly, they were able to move fluidly between identifying as mestizo (mixed race) and as Black.

One of the participants in a non-Afro focus group, a dark-skinned woman, self-identified as mestiza during the recruitment phone call. When she was asked her racial identification during the focus group—this time in a room of seven light skinned mestizos—she proudly claimed, "I am Afro-Panamanian". Many Afro-Panamanians recognize that they are both mixed and Black. Their ability to claim race-mixture does not necessarily negate Black self-recognition (Johnson 2020, p. 373).

### 5. Positive Afro-Latin@ Representation and Afro-Latin@ Youth Identity

Thus far, this essay has discussed the complexities associated with Afro-Latin@ identity, the historical manifestations of anti-Blackness in Latin America and beyond, and how mestizaje and colorism function within Afro-Latin@ communities. The discourse around mestizaje, blanqueamiento and Blackness in the Latin@ community highlights the importance of positive Afro-Latin@ representation in literature to affirm, reshape, and combat negative perceptions of Blackness and Black identity. The historical context presented frames this argument of how positive Afro-Latin@ representation within youth and children's literature can help Afro-Latin@ young people embrace the positive attributes of their African identities. As Jank's suggests, it is essential for Latin@s from different cultural backgrounds to have the ability to reshape existing discourses (qtd. in Braden and Rodriguez 2016, p. 58). Braden and Rodriguez (2016) posit that the presence of multicultural literature provides a window into understanding the culture and experiences of others. Children who are presented with literature that reflects a singular cultural identity or heritage tend to place that cultural experience above all others (Braden and Rodriguez 2016). This, of course, creates a climate where there is disbelief of cultural significance on the part of young people of color. The authors emphasize the importance of children's literature being inclusive in schools in helping to affirm self-identity and allowing children to begin to resist the negative ways some groups are represented through storytelling (Braden and Rodriguez 2016).

The possibility of literature as a tool of resistance is demonstrated by Xie (2013) who pointed to children's literature implemented as a postcolonial tool to decolonize and deconstruct racial and ethnic differences in an attempt to move toward "globalized post-coloniality" (p. 13). Durand and Jiménez-García (2018) support Xie's anti-colonial perspective in their discussion of visual narratives and their potential to deconstruct colonial ideas of race. The authors mention Eric Velasquez, an author challenging homogenous beliefs about African American, Afro-Caribbean, and Afro-Latin@ people. Through his work and visual representations, Velasquez challenges existing depictions of Afro-Latin@ and Afro-Caribbean people by presenting alternate images and ideas of Black identity (Durand and Jiménez-García 2018).

When considering the role positive Afro-Latin@ representation plays in working to affirm Afro-Latin@ cultural identity, one must understand how children generally form ideas surrounding racial identity. Children generate perceptions of race at early ages, and within the US even if not actively involved in discussions of race, they are inextricably forced to contend with the systemic aftermath of its social construction. For example, young children of color often live in segregated neighborhoods, and in school and beyond, they are subjected to literature and media that actively center white lead characters and highlight their lived experiences (White and Wanless 2019). Additionally, children, early on, comprehend the race-based social order within the United States that positions and privileges some racial groups above others. This understanding points to the importance of presenting Afro-Latin@ children with positive cultural representations to prepare them to contend with the inevitable systemic racism they will undoubtedly encounter (White and Wanless 2019). It is equally important to note that schools often function as sites of cultural and racial erasure as children of color seldom see themselves represented or reflected in the books available to them in schools, and this lack of representation often works to negatively impact their perceptions of self (Pérez Huber et al. 2023).

Mizell (2022), in his work, emphasizes the power associated with children being able to access literature reflective of their cultural identity. Using LatCrit and testimonios, he explores the immigrant story of a young Afro-Latin@ boy. Latin@ Critical Studies, or LatCrit, is an offshoot of Critical Race Theory and emphasizes the usage of testimonio[2] and counter-storytelling powerful tools to reconcile historical wrongs by countering singular white colonist narratives (Lopez 2007; Solórzano and Yasso 2001; Tran 2019) Mizell (2022), by emphasizing the potential of counter-storytelling in picture books, explores the possibility of picture books being used as an entry-point to provide students of color

with counternarratives decentering whiteness and hegemony and combating anti-Latin@ and anti-Black perspectives. Mizell (2022) contends that when presented in picture book format, "testimonios reinforce the lived experiences of Latine immigrant children to contest dominant deficient or inaccurate narratives and build a shared emphatic understanding with others" (p. 6).

## 6. Shifting the Narrative: Racial Representation and Affirming Cultural Identity

In a general search of Afro-Latin@ children's books, this author encountered an article published by Essence magazine, a popular Black women's magazine, depicting seventeen children's books featuring Afro-Latin@ and Afro-Caribbean characters (Peart 2020). Many books were written and illustrated by Afro-Caribbean/Afro-Latin@ authors and illustrators. Some books were self-published by the authors, some through publishers outside of the US, and others by smaller divisions of larger, well-known publishers. Smaller, less-known publishers published most of the titles. Notably, Harper Collins, a larger-known publishing house, published a few of the titles. The books listed were mainly for younger children; however, the article mentions a few young adult novels. Broadly, the books in the article address topics relevant to young Afro-Latin@ readers, such as colorism and navigating dual cultural identities.

The following titles are presented in the article and reviewed independently by this author: *Bad Hair Does Not Exist*, which was written and illustrated by Sulma Arzu-Brown, a Garifuna-Honduran author and illustrator, and published by Afro-Latin Publishing, a company dedicated to publishing titles that focus on the experiences of Afro-descendants. Aruzu-Brown's work attempts to combat the colonial narrative commonly reproduced throughout Latin America around the existence of "good" hair (straight and silky) and "bad" hair (coarse or tightly coiled); *Cendrillon: A Caribbean Cinderella* by Robert D. San Souci, a white San Francisco-based writer known for reimagining global folktales, illustrated by Brian Pickney, an award-winning African-American artist, and published by Aladdin Books, a division of Simon and Shuster dedicated to publishing culturally relevant books. The book presents colorful images that help to paint a picture of island life. *Down by the River: Afro-Caribbean Rhymes, Games, and Songs for Children* by Grace Hallworth, a Trinidadian author, illustrated by Caroline Binch, a white illustrator, and published by a U.K.-based publisher, Frances Lincoln Children's Books. Hallworth exposes children to popular Caribbean children's songs and games. Her work helps to showcase the rich history of storytelling in Caribbean cultures; *Drum Dream Girl* by Margarita Engle, a Cuban-American author, published by Harper Collins and illustrated by Mexican-born artist Rafael Lopez, is based upon the childhood of Millo Castro Zaldarriaga, a Chinese-African-Cuban musician who fulfilled her childhood dream of becoming a drummer, a profession traditionally frowned upon for women in Cuba. Millo later would form the first all-female dance band, Anaconda, with her sisters. *Isabella's Hair and How She Learned To Love It* by Brooklyn-based English as a Second Language (ESL) teacher and Afro-Latin@ author Marshalla Soriano Ramos and illustrated by Michael Murphy, a children's book illustrator. Ramos self-published the book through CreateSpace Independent Publishing Platform, an Amazon publishing service. The illustrations are crudely depicted and not as stylized as in other reviewed books. However, the story highlights the experiences of an Afro-Puerto Rican child and her struggle to accept her natural hair. *Kitchen Dance*, written and illustrated by Maurie J. Manning, a white-presenting author and published by Harper Collins, presents a positive image of an Afro-Latin family who enjoys singing and spending time with each other. *Letters to My Mother* by Teresa Cárdenas, an Afro-Cuban author, was published by Groundwood Books, a Canadian-based publisher known for publishing books by Latin@ authors in English and Spanish. The book is a young adult novel featuring an Afro-Cuban girl who moves in with her aunt and cousins after her mother's death. The main character endures attacks from her family related to her physical features (hair, dark, and skin color). To deal with her suffering, she writes letters to her deceased mother. The story helps to illustrate the colorism within families and its damaging effects on individuals. *Marisol and*

*Magdalena* is a young adult novel written by Afro-Panamanian author Veronica Chambers and published by Little Brown Books for Young Readers. The story depicts the experiences of two Panamanian-American girls sent to Panama to live with their grandmother for a year. The book is a coming-of-age story highlighting the uniqueness of being an American-born Panamanian who has to navigate life in Panama. *Max Loves Muñecas*! by Zetta Elliott, a Black Canadian author, illustrated by Mauricio Flores, a Honduran freelance illustrator, and published by CreateSpace, tells the story of young Max, a boy who likes dolls. The author, through her storytelling, attempts to deconstruct and challenge the notion of traditional gender roles; *My Feet Are Laughing* by Lissette Norman, Afro-Dominican poet and writer, illustrated by Frank Morrison, an African-American graffiti artist and former break-dancer, and published by the American publishing company, Farrar, Straus and Giroux. The story focuses on a young, out-spoken Afro-Latin@ girl who moves with her mom and sister from her family apartment to her maternal grandmother's New York City brownstone following her grandmother's death. The book incorporates poetry into the storytelling and presents a non-traditional depiction of family as the main character's parents are amicably divorced and committed co-parents, showing that divorce does not mean the complete dissolution of the family unit; *Me Llamo Celia* by Monica Brown, a Peruvian-American author, illustrated by Rafael Lopez (mentioned previously, he also illustrated *Drum Dream Girl*), and published by Cooper Square Publishing, LLC. The book is colorful and beautifully illustrated. It introduces young children to the life of the renowned Afro-Cuban singer Celia Cruz; *Niña Bonita* by Ana Maria Machado, a Brazilian-born writer, illustrated by Venezuelan illustrator Rosana Feria and published by Kane/Miller Book Publishers. The book tells the story of a dark-skinned Afro-Latin@ girl who meets a white bunny who loves her dark skin and wants to be black and beautiful like her. The book explores colorism within families as the main character is the darkest member of her family; *Pelé, El Rey del Fútbol* by Monica Brown, Peruvian American author, illustrated by Rudy Guiterrez and published by Harper Collins. The book focuses on the life of the famous Afro-Brazilian footballer Pelé and his rise to fame. The illustrations attempt to capture the motion and smooth artistry of Pelé's footwork on the soccer field. *Secret Saturdays* is a fictional novel for young adults written by Afro-Puerto Rican author Torrey Maldonado, a Brooklyn public school teacher. The story captures a friendship between two young Afro-Latin@ boys raised by single mothers and growing up in the Brooklyn Red Hook Project, where the author grew up. *Shadowshaper* by Daniel José Older, a Cuban-American author and published by Scholastic Inc, is a young adult novel about a young Afro-Latin@ girl searching for her Afro-Boricua roots. Older tends to write young adult stories that center the experiences of multicultural/ethnic characters. *Show and Prove* is a young adult novel by Sofia Quintero, a Bronx-raised Afro-Puerto Rican-Dominican author, and published by Knopf Books for Young Readers, a division of Penguin Random House Canada. The book is set in the Bronx, NY, during the early 1980s and deals with the racial tensions that arose during the early days of hip hop; finally, *A Song for Bijou* is a novel written by Josh Farrar, a white Brooklyn-based author, and published by Bloomsbury USA Children's. The book features a Haitian girl who has recently immigrated to the US. The young girl meets a white boy interested in dating her, and the book chronicles the cultural challenges associated with intercultural dating. The author is not a member of Haitian culture, so it is unclear how closely he captured the Haitian experience; however, his bio states that he interviewed members of the Haitian community before writing his book (Peart 2020).

This preliminary search highlights some of the existing literature that positively represents the Afro-Latin@–Afro-Caribbean experience. Still, despite the existence of these books and progress in children's literature, growth has hovered between 13 and 15 percent from 2002 to 2015 and has not exceeded 15 percent since then (Nel 2017). Nel (2017) attributes this stagnant growth to racism within the children's publishing industry and attempts on the part of publishers to maintain institutional racism through the implementation of "colorblind" practices that effectively deny the marketability of works that center on the experiences of people of color. So, given the structural racism and institutional racism that

exists in the US, how do we go about ensuring the creation and publication of books that feature positive representations of Afro-Latin@ characters? Nel (2017) suggests that one way to combat the colorblind practices present in the children's publishing field that aim to white-wash Black characters and to counter the resistance to telling Afro-Latin@ stories on a larger scale is to increase the number of editors of color in these largely white spaces who are willing to publish diverse stories.

Additionally, some of the institutional racist practices in children's publishing are now being combated on an individual level by authors such as Zetta Elliott, who has decidedly made it a point to cut out the middle person and self-publish her work to ensure that her multicultural stories continue to be told (Nel 2017). Elliott (2015) identifies herself as a Black feminist writer who writes stories about Black children and teens. She spoke of her experiences as a writer of youth literature and the challenges associated with attempting to publish her work through traditional methods:

Since I started writing for young readers in 2000, only three of my thirty stories have been published traditionally. I turned to self-publishing as my only recourse, and now face the contempt of those who see self-publishing as a mere exercise in vanity (Elliott 2015, p. 1).

Elliott (2015) expresses that her reasons for moving towards self-publishing were largely to combat traditional publishing ideas about the perceived benefits of publishing stories featuring people of color. Additionally, she speaks of self-publishing as a means to combat the white privilege present in the field, self-publishing then serving as a means for multicultural stories to be told without the permission of the largely white-dominated publishing industry (Elliott 2015).

## 7. Final Thoughts

Today's highly politicized climate evidences a cultural shift away from an attempt to celebrate the richness and diversity of lived experience in the United States to one aimed toward suppressing marginalized voices and forms of expression, literary work among them (Lowery 2023). Now more than ever, those committed to social justice, dismantling white supremacy and hegemony, must be ever vigilant of these attempts to ban the distribution of knowledge that falls outside the parameters of societal white norms. As mentioned throughout this essay, the presence of positive literary examples plays a role in affirming a child's cultural identity. Multicultural children's authors committed to inclusive storytelling and increasing Afro-Latin@ presence in children's literature must begin to think creatively about how to combat the anti-progressive pushes happening all across our country and around the world.

The tendency on the part of traditional children's publishing houses to dismiss or minimize the need for inclusive literature and the telling of non-white stories must be challenged, but in the words of Audre Lorde, "the master's tools will never dismantle the master's house" and given our understanding of the way that white supremacy operates in our society, writers of color must begin to think creatively about how to distribute their work to the masses. If book banning continues and if white-run publishing houses continue to discredit and minimize the importance of non-white voices, self-publishing may be a means to do so. Regardless of the approach, as academics, we must challenge the current political discourse that aims to discredit or diminish the contributions of non-white Americans. Diverse stories deserve to be told, Afro-Latin@ stories among them. Afro-Latin@ children have a right to see themselves reflected in children's literature and their cultural identities affirmed, and we must collectively work to ensure that happens.

**Funding:** This research received no external funding.

**Data Availability Statement:** The original contributions presented in the study are included in the article, further inquiries can be directed to the corresponding authors.

**Conflicts of Interest:** The author declares no conflict of interest.

## Notes

[1]    "Black" and "Latin@" will be capitalized in this work, while "white" will remain lowercase. This grammatical decision is a purposeful attempt by the writer to decenter whiteness and white people in the discussion (Appiah 2020). In addition, Latin@ is used instead of "Latine", "Latinx" or "Latino/a" to affirm and promote gender neutrality (Lopez Torregrosa 2021). Despite recent academic pushes towards using *Latinx*, as an Afro-Latin@, this writer does not prefer it.

[2]    According to Rodriguez-Campo (2021), "Testimonio involves bearing witness to the collective experiences of historically marginalized communities, particularly as it relates to their oppression, resistance, and resilience. As an approach, it is an inherently decolonial process since it decenters Eurocentric knowledge and challenges power. Unlike oral history, memoir, or autoethnography, testimonio positions itself as an urgent and political voicing that rejects notions of objectivity and neutrality".

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
