# Peer review of "Afro-Latin@ Representation in Youth Literature: Affirming Afro-Latin@ Cultural Identity"

_humanities, doi:10.3390/h13010027_

Round 1
Reviewer 1 Report
Comments and Suggestions for Authors
This is an excellent article. I enjoyed reading it. Your sources are perfect, and the topic is of utmost importance today.

There are a few minor suggestions with typos that I have highlighted in the text with comments. In the works cited section, I would highly recommend placing titles of articles in quotation marks and capitalizing major words (nouns, pronouns, adjectives, verbs, adverbs, and subordinate conjunctions). The titles appear to be cited using the rules in Spanish with only the first letter and first letter after the colon capitalized.
Author Response
Thank you so much for your edits. Enclosed you will find my response.
Regards,

Reviewer 2 Report
Comments and Suggestions for Authors
The author provides a thorough review of the literature of children, identity, and representation in children’s literature. The author also covers the complexity of entities affecting children of color. The discussion on colorism is quite good. Additionally, the author has a timely response to the myths perpetuated in the publishing industry regarding minority literature.
Additionally, the exploration of the titles found in Essence Magazine is useful. A logical next step for the author or future researchers would be an analysis of what children’s books and young adult novels say about culture and identity. The article works as a purse informational piece.
Check lines 311-312 and 393 to repair awkward syntax.
There is no central question addressed, rather the article reinforces the need to have minority students represented in children’s literature. It then surveys a list of titles found in Essence Magazine. The article itself is interesting but it does not take on any pressing questions in the field of study. Also, I was surprised to see AfrioLatin@--which was largely used in the 1980s and has since been supplanted by Latinx.
Comments on the Quality of English Language
Moderate editing of English language required.
Author Response
Hi Iva, I found the Word version and my edits are enclosed (I kept track changes on the document). Please let me know if anything else is needed.
